# FastFit: Accelerating Multi-Reference Virtual Try-On via Cacheable Diffusion Models

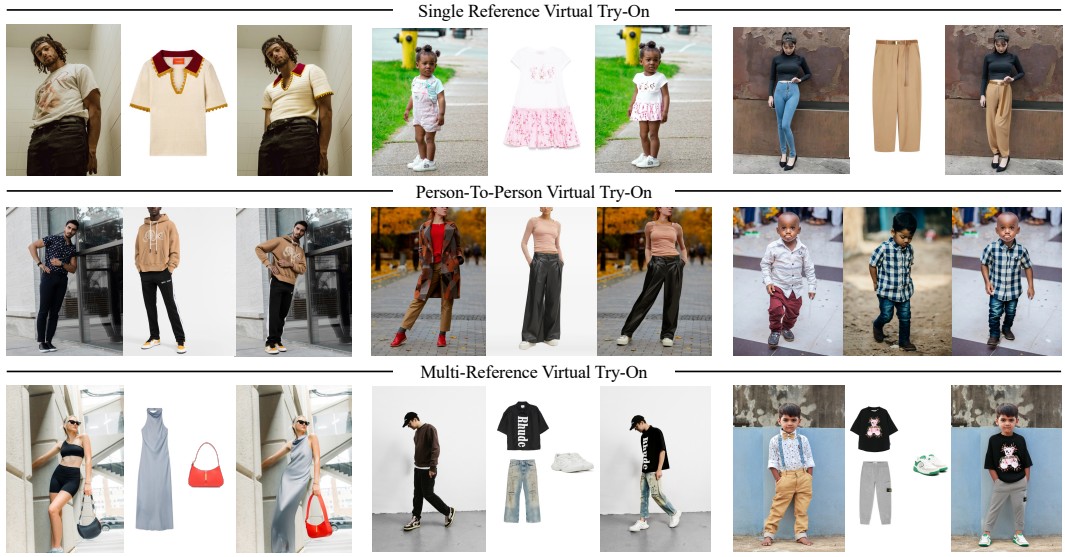

Figure 1: FastFit provides an accelerated solution for diverse virtual try-on tasks, including single-reference, person-to-person, and our primary focus, multi-reference composition. By decoupling the reference images from the denoising process, our cacheable diffusion architecture delivers high-fidelity virtual try-on across multiple challenging scenarios at a much faster speed.

## ABSTRACT

Despite its great potential, virtual try-on technology is hindered from real-world application by two major challenges: the inability of current methods to support multi-reference outfit compositions (including garments and accessories), and their significant inefficiency stemming from the redundant re-computation of reference features during denoising. To address these challenges, we propose FastFit, a high-speed multi-reference virtual try-on framework based on a novel cacheable diffusion architecture. By combining the Semi-Attention mechanism and substituting traditional timestep embeddings with class embeddings for reference items, our model fully decouples reference feature encoding from the denoising process with negligible parameter overhead. This allows reference features to be computed only once and losslessly reused across all steps, fundamentally breaking the efficiency bottleneck and achieving an average $3.5\times$ speedup over comparable methods. Furthermore, to facilitate research on complex, multi-reference virtual try-on, we introduce DressCode-MR, a new large-scale dataset. It comprises 28,179 sets of high-quality, paired images covering five key categories (tops, bottoms, dresses, shoes, and bags), constructed through a pipeline of expert models and human feedback refinement. Extensive experiments on the VITON-HD, DressCode, and DressCode-MR datasets show that FastFit surpasses state-of-the-art methods on key fidelity metrics while offering its significant advantage in inference efficiency.

# 1 INTRODUCTION

Generative AI-based virtual try-on has recently made remarkable progress. An ideal virtual try-on system—one that could revolutionize online retail and power applications like intelligent outfit visualization—would allow users to seamlessly mix and match various garments and accessories, rapidly generating photorealistic results to enable an interactive experience. However, two major challenges hinder current methods from achieving this vision. Firstly, most existing methods (Xie et al., 2022; Wang et al., 2018; Xu et al., 2024; Choi et al., 2024; Chong et al., 2024; Jiang et al., 2024) are designed for a single reference garment (e.g., a top or a dress), requiring a complete multi-item outfit to be rendered through iterative passes, leading to both inflated computation time and the risk of accumulated synthesis errors. Furthermore, the general lack of support for essential accessories like shoes and bags prevents the generation of truly holistic and realistic outfits. Secondly, the computational inefficiency of current methods stems from two competing yet flawed strategies, as illustrated in Figure 2. On the other hand, in-context learning-based methods (Guo et al., 2025; Chong et al., 2024; Huang et al., 2024a) repeatedly process the concatenated reference and person features at each of the $N$ denoising steps (Figure 2 (b)), causing significant computational redundancy due to the reprocessing of static reference

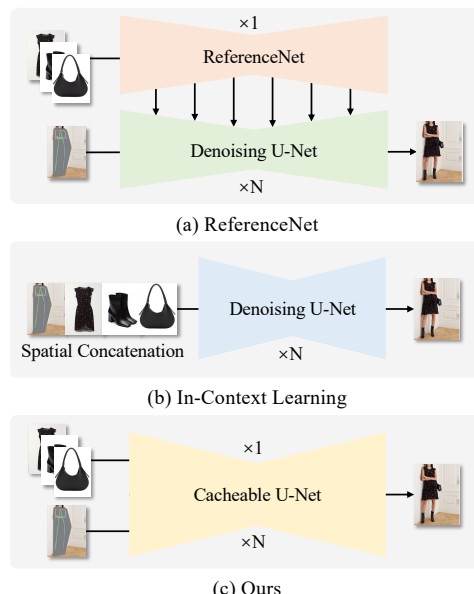

(a) ReferenceNet

(b) In-Context Learning

(c) Ours

Figure 2: Architectural comparison of multi-reference try-on methods. Our cacheable U-Net (c) avoids the parameter overhead of ReferenceNet (a) and the computational redundancy of In-Context Learning (b).

features. On the other hand, ReferenceNet-based methods (Huang et al., 2024b; Choi et al., 2024; Xu et al., 2024; Zhang et al., 2024b; Zhou et al., 2024; Jiang et al., 2024) employ a separate network to encode references (Figure 2 (a)). While this explicit separation avoids the aforementioned computational redundancy, it introduces substantial parameter overhead, increasing both training and inference costs.

To overcome these limitations, we introduce FastFit, a high-speed framework that enables coherent multi-reference virtual try-on through a novel cacheable diffusion architecture. Our proposed Cacheable UNet decouples the reference feature encoding from the iterative denoising process, which is achieved by introducing a Reference Class Embedding and a Semi-Attention mechanism. This structure enables a Reference KV Cache during inference, which allows reference features to be computed only once and losslessly reused in all subsequent steps, fundamentally breaking the efficiency bottleneck and achieving an average 3.5× speedup over comparable methods with negligible parameter overhead. Furthermore, observing the lack of datasets with complete outfit pairings, we construct DressCode-MR, a large-scale multi-reference try-on dataset based on Morelli et al. (2022). We developed a data-generation pipeline that trains expert models based on Chong et al. (2024) and Labs (2024) to recover canonical images of individual items, and utilizes human feedback to ensure high quality. This results in 28,179 multi-reference image sets spanning five key categories: tops, bottoms, dresses, shoes, and bags.

In summary, the contributions of this work include:

- We propose FastFit, a novel framework for high-speed, multi-reference virtual try-on. It is the first to enable coherent multi-reference virtual try-on across five key categories, including tops, bottoms, dresses, shoes, and bags, while achieving an average 3.5× speedup over comparable methods.

- We design a novel Cacheable UNet structure featuring a Reference Class Embedding and a Semi-Attention mechanism. This design decouples reference feature encoding from the denoising process, enabling a lossless Reference KV Cache that breaks the core efficiency bottleneck of subject-driven generation architectures.

Figure 3: Illustrative examples from our proposed DressCode-MR dataset. Each sample provides a pairing of a full-body person image with a set of corresponding canonical images for each item.

- We construct DressCode-MR, the first large-scale dataset specifically for multi-reference virtual try-on. It comprises 28,179 high-quality image sets, providing a solid foundation to foster future research in complex outfit generation.
- We conduct extensive experiments on VITON-HD, DressCode, and DressCode-MR datasets, demonstrating that FastFit surpasses state-of-the-art methods in image fidelity while maintaining its significant efficiency advantage.

## 2 RELATED WORK

### 2.1 SUBJECT-DRIVEN IMAGE GENERATION

To enable finer-grained control, subject-driven image generation has become a key research focus. Early efforts primarily centered on single reference images, injecting specific subject identities or artistic styles by fine-tuning model weights (Ruiz et al., 2022; Yang et al., 2022; Hu et al., 2021; Huang et al., 2024a) or utilizing lightweight adapters (Ye et al., 2023; Mou et al., 2023; Chen et al., 2023). However, the former approach requires training a separate model for each subject, limiting its practical flexibility, while the latter, despite being convenient, often faces challenges in maintaining high fidelity to the reference image. Some methods based on in-context learning, such as IC-LoRA (Huang et al., 2024a) and OminiControl (Tan et al., 2025a), achieve superior detail preservation by concatenating the reference image with noise along the spatial dimension. The trade-off is that the reference must participate in every denoising step, which is the root cause for significantly increasing inference time and computational cost. EasyControl (Zhang et al., 2025) and OminiControl2 (Tan et al., 2025b) achieve further acceleration through feature reuse by adjusting the attention map. However, they require creating additional trainable branches for each condition. This architectural dependency on separate branches makes scaling to multiple references cumbersome and computationally expensive. Consequently, some works have begun to explore multi-reference generation; for instance, IC-Custom (Li et al., 2025) inputs multiple images as a single concatenated map for multi-concept composition, and MultiRef (Chen et al., 2025) provides a systematic definition and benchmark for multi-reference generation. Nevertheless, in the domain of virtual try-on, multi-reference generation remains an under-explored area. How to harmoniously compose visual information from multiple references while mitigating the heightened computational or parameter burden from increased inputs remains a significant and open challenge.

### 2.2 IMAGE-BASED VIRTUAL TRY-ON

Image-based virtual try-on aims to realistically synthesize a person wearing target garments. Classic paradigms centered on a warp-and-fuse method, which explicitly deforms the garment using either geometric transformations or learned appearance flows before the blending stage (Wang et al., 2018; Han et al., 2018; Choi et al., 2021; Han et al., 2019; Ge et al., 2021; Xie et al., 2021; 2023; Gou et al., 2023; Chong & Mo, 2022); however, these approaches are frequently hampered by visual artifacts from inaccurate warping. Subsequently, the advent of diffusion models revolutionized the field by reframing the task as end-to-end conditional image generation, bypassing the error-prone warping step. The dominant strategy in these modern models involves injecting high-fidelity garment features into the denoising process via sophisticated conditioning mechanisms, such as parallel encoder branches (i.e., ReferenceNets) or ControlNet(Zhang et al., 2023)-like structures, a technique employed by a vast body of recent work (Zhu et al., 2023; Morelli et al., 2023; Kim et al., 2023; Xu et al., 2024; Wang et al., 2024; Choi et al., 2024; Sun et al., 2024; Zhou et al., 2024; Zhang et al., 2024a; Kim et al., 2024). Recent innovations further push the boundaries by exploring alternative backbones like Diffusion Transformers (Peebles & Xie, 2022) or introducing novel control

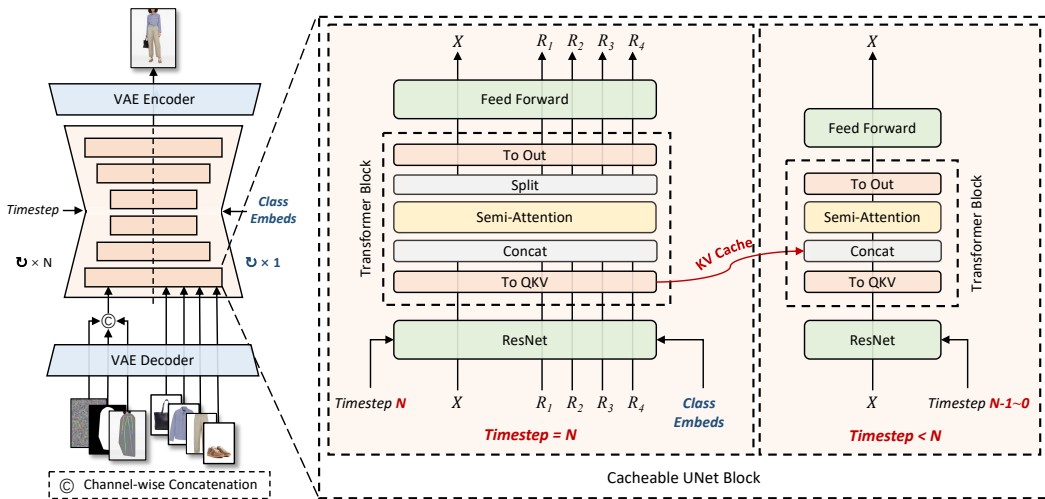

Figure 4: Overview of the FastFit Inference Pipeline. The left side depicts the overall flow, consisting of a VAE Encoder, a Cacheable UNet, and a VAE Decoder. The input to the UNet, denoted as $X$, is a concatenation of the noisy latent $z_n$ and person conditions $c_p$. The right side details the proposed Cacheable UNet Block. This block operates in two modes: at the initial timestep $N$, it computes and caches features from reference inputs $R_1 \ldots R_4$; at subsequent timesteps ($t < N$), it reuses these cached features via Semi-Attention to accelerate the denoising process.

modalities such as textual prompts and more generalized conditioning schemes (Guo et al., 2025; Jiang et al., 2024). Despite achieving unprecedented realism, their inference speed and general limitation to single garments have become key bottlenecks, hindering the technology's application in real-world scenarios that demand rapid feedback and multi-item outfit composition.

## 3 METHODS

### 3.1 OVERVIEW

The overall framework of FastFit is built upon the foundation of Latent Diffusion Models (LDMs) (Rombach et al., 2021) and is designed to achieve high-speed, multi-reference virtual try-on through a novel conditioning cacheable UNet architecture. To clarify the data flow during inference, the entire pipeline is depicted in Figure 4, with the detailed mechanism of our core component shown on the right side. To ensure the generated image preserves the person's identity and pose while accurately rendering the new garments, we prepare two sets of conditions:

**Person Conditioning** $c_p$: To accurately preserve the person's identity and body pose, we construct the person condition $c_p$. First, we utilize AutoMask (Chong et al., 2024) to generate a cloth-agnostic mask $\mathbf{M_a}$ from the input image $I_p$. Subsequently, a composite image, $\mathbf{I_{comp}}$, is created by combining the human pose skeleton extracted via DWPose (Yang et al., 2023) with the person image masked by $M_a$. $c_p$ is formed as:

$$c_p = \text{Concat}(\text{Interpolate}(M_a), \mathcal{E}(I_{comp})) \tag{1}$$

where $\mathcal{E}$ is the VAE encoder, Interpolate is a downsampling function that resizes the mask $M_a$, and Concat denotes the channel-wise concatenation. Finally, the main input to the UNet, denoted as $X$ in Figure 4, is obtained by concatenating the person condition $c_p$ and the Gaussian noise $z_N$ along the channel dimension:

$$X = \text{Concat}(z_N, c_p) \tag{2}$$

**Reference Conditioning** $\{R_i\}_{i=1}^K$: To capture the detailed appearance of the target garments, we extract a set of reference latents $\{R_i\}_{i=1}^K$ from the corresponding reference images $\{I_{R_i}\}_{i=1}^K$, which is defined as:

$$\{R_i\}_{i=1}^K = \{\mathcal{E}(I_{R_i})\}_{i=1}^K \tag{3}$$

In the detailed block diagram of Figure 4, we illustrate this with $K = 4$ examples ($R_1 \ldots R_4$) for visual clarity.

The image generation process is guided by a denoising UNet $\epsilon_\theta$, which predicts the noise $\tilde{\epsilon}_t$ at each timestep $t$. As shown in the right panel of Figure 4, we replace standard UNet blocks with our **Cacheable UNet Blocks** to enable efficient multi-reference processing. our key innovation is to conceptually partition the function of $\epsilon_\theta$ into two streams: a time-independent path for reference inputs and a time-dependent path for the denoising process.

**Phase 1: Caching at Timestep $N$.** As illustrated in the 'Timestep $N$' path of the block diagram, each reference latent $R_i$ is processed individually along with the denoising input $X$, conditioned on its corresponding Class Embedding $E_i$. This allows us to compute and cache a separate feature representation, $\mathcal{R}^{(i)}_{\text{cache}}$, for each item. This operation is performed for all $i \in \{1, \ldots, K\}$:

$$\mathcal{R}^{(i)}_{\text{cache}} = \epsilon_\theta(R_i, E_i) \quad \text{for } i = 1, \ldots, K \tag{4}$$

The resulting set of cached features, $\{\mathcal{R}^{(i)}_{\text{cache}}\}_{i=1}^K$, is stored for reuse.

**Phase 2: Reuse at Timestep $t < N$.** For the subsequent denoising steps $t = N - 1 \ldots 0$, as shown in the 'Timestep $N - 1 \sim 0$' path, the UNet only processes the time-dependent input $X$. It integrates the static reference information by reading the pre-computed set of cached features, $\{\mathcal{R}^{(i)}_{\text{cache}}\}_{i=1}^K$, via a Semi-Attention mechanism (detailed in Section 3.2):

$$\tilde{\epsilon}_t = \epsilon_\theta(z_t, c_p, \gamma(t), \{\mathcal{R}^{(i)}_{\text{cache}}\}_{i=1}^K) \tag{5}$$

This decomposition of the denoising process is the key to FastFit's efficiency, as it shifts the expensive computation for multiple reference features entirely out of the iterative loop. Once the process concludes at $t = 0$, the final clean latent, $z_0$, is mapped back to the pixel space using the VAE decoder $\mathcal{D}$, to produce the high-resolution output image, $I_{\text{out}}$:

$$I_{\text{out}} = \mathcal{D}(z_0) \tag{6}$$

### 3.2 CACHEABLE UNET FOR EFFICIENT CONDITIONING

The primary bottleneck in existing subject-driven diffusion models is the repeated computation of reference features at every denoising step. This is because the reference conditioning is typically dependent on the timestep $t$, making the features dynamic. Our key innovation, the Cacheable UNet, fundamentally breaks this dependency, enabling reference features to be computed once and reused. This is achieved through two core components: Reference Class Embedding and a Semi-Attention mechanism, as illustrated in Figure 4 (b).

**Reference Class Embedding.** To decouple the reference features from the denoising timestep $t$, we replace the conventional timestep embedding with a static, learnable Reference Class Embedding for the reference items. Specifically, for a set of $K$ reference items $\{R_1, \ldots, R_K\}$, each belonging to a certain category (e.g., 'top', 'shoes'), we introduce a corresponding set of learnable class embeddings $\{E_1, \ldots, E_K\}$. The features for each reference item $R_i$ are conditioned on its class embedding $E_i$ instead of the shared timestep embedding $\gamma(t)$ used by the denoising features $X$. The reference class embedding is injected in the same manner as the timestep embedding; both modulate the features within the ResNet blocks through an scaling operation. Since the class embeddings are constant throughout the entire denoising process, the resulting reference features become static and independent of the current timestep $t$, making them inherently cacheable.

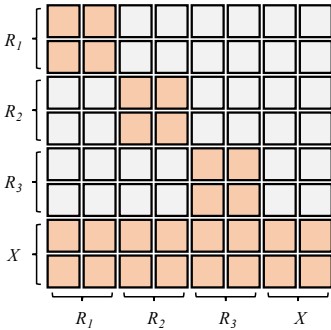

Figure 5: Visualization of the Semi-Attention Mask. Denosing feature $X$ attend to all features, while each reference feature $R_i$ is restricted to its own.

**Semi-Attention Mechanism.** Having made reference features static, we need a mechanism to inject their information into the denoising process without compromising their static nature. A standard full self-attention would allow information to flow from the step-dependent denoising features $X$ back to the reference features $R_i$, thereby "contaminating" them and breaking the condition for caching. To solve this, we propose a Semi-Attention mechanism, visualized in Figure 5. In this design, we treat both the denoising features $X$ and all

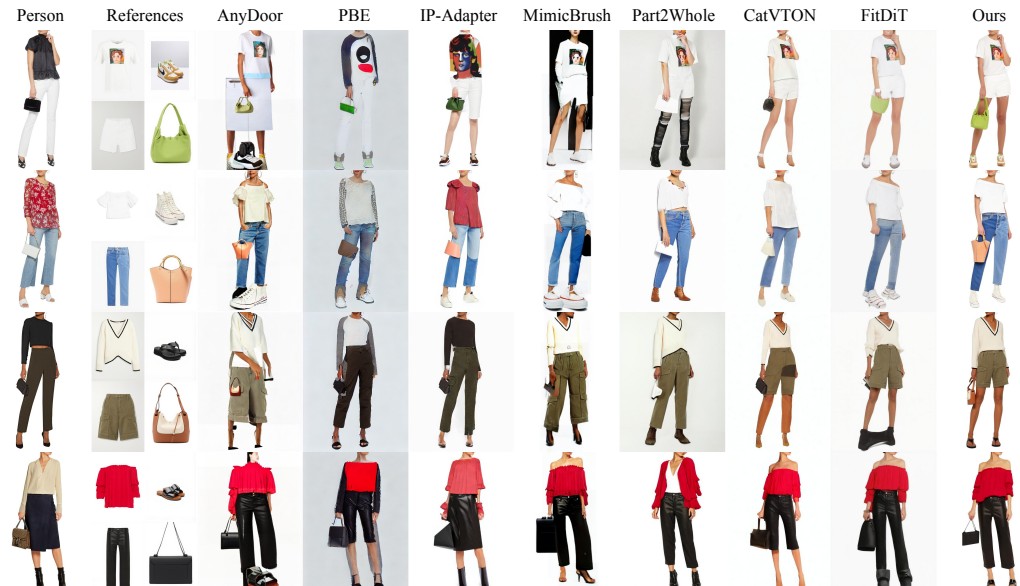

Figure 6: Qualitative comparison on the DressCode-MR dataset. See Section 5 for more results. Please zoom in for more details.

reference features $\{R_i\}$ as a single sequence of tokens. The attention calculation is governed by a specific mask that controls the information flow: **(1) Denoising-to-All:** The tokens of the denoising feature $X$ are allowed to attend to all tokens in the sequence (i.e., to itself and to all reference features $R_i$). This allows the model to effectively "read" the appearance information from each garment and apply it to the person. **(2) Reference-to-Self:** The tokens of each reference feature $R_i$ are only allowed to attend to themselves. They cannot attend to the denoising features $X$ or to any other reference feature $R_j$ (where $j \neq i$). This attention mask ensures that the reference features act as a static, read-only source of information for the denoising process. Their representations are never updated by the dynamic features of $X$, thus preserving their cacheability across all timesteps.

OminiControl2 (Tan et al., 2025b) and EasyControl (Zhang et al., 2025) also employ a semi-attention-like mechanism for KV caching. However, their reliance on dedicated trainable branches and treat reference encoding statically by fixing the timestep to zero, thus failing to utilize this input slot for fine-grained control. Our Cacheable UNet addresses this by replacing the uninformative zero-timestep embedding with informative class embeddings, which allows the a single model to explicitly attend to specific garment types or regions based on the input category, as demonstrated by the ablation results in Fig. 9.

### 3.3 INFERENCE ACCELERATION WITH REFERENCE KV CACHE

The design of our Cacheable UNet enables a highly efficient inference pipeline via a Reference KV Cache. As depicted in Figure 4(b), the process is split into two stages:

**Pre-computation and Caching (One-time Cost).** Before the iterative denoising loop begins, we perform a single forward pass for each reference item $R_i$ through the UNet $\epsilon_\theta$. For each Semi-Attention layer, we then compute and store its corresponding Key ($K_i^{\text{cache}}$) and Value ($V_i^{\text{cache}}$) matrices. This pre-computation step is performed only once per generation request.

**Accelerated Denoising Loop.** For every subsequent denoising step from $t = N - 1$ down to 0, we completely bypass the computation for the reference branches. Instead, for each Semi-Attention layer, we only compute the Query ($Q_X$), Key ($K_X$), and Value ($V_X$) from the current denoising features $X_t$. We then construct the full key and value matrices, $K_{\text{full}}$ and $V_{\text{full}}$, by concatenating these dynamic tensors with all the cached keys $\{K_i^{\text{cache}}\}_{i=1}^K$ and values $\{V_i^{\text{cache}}\}_{i=1}^K$, respectively:

$$K_{\text{full}} = \text{Concat}(K_X, K_1^{\text{cache}}, \ldots, K_K^{\text{cache}}) \tag{7}$$

$$V_{\text{full}} = \text{Concat}(V_X, V_1^{\text{cache}}, \ldots, V_K^{\text{cache}}) \tag{8}$$

Figure 7: Qualitative comparison on the VITON-HD (Choi et al., 2021) and DressCode (Morelli et al., 2022) dataset. See Section 5 for more results. Please zoom in for more details.

The final attention output is then calculated only for the denoising query $Q_X$:

$$\text{Attention}(Q_X, K_{\text{full}}, V_{\text{full}}) = \text{softmax}\left(\frac{Q_X K_{\text{full}}^T}{\sqrt{d_k}}\right) V_{\text{full}} \tag{9}$$

This strategy effectively reduces the computational cost of attention at each step to be dependent only on the denoising features, regardless of the number or complexity of reference items. This fundamentally resolves the efficiency bottleneck, leading to a substantial reduction in inference latency, especially in the multi-reference setting central to our work.

## 4 EXPERIMENTS

### 4.1 DATASETS

We evaluate our model on three datasets, VITON-HD (Choi et al., 2021), DressCode (Morelli et al., 2022), and our newly proposed DressCode-MR, all at 1024×768 resolution. VITON-HD (Choi et al., 2021) provides 13,679 image pairs for upper-body virtual try-on (11,647 train / 2,032 test). DressCode (Morelli et al., 2022) dataset features 53,792 full-body pairs (48,392 train / 5,400 test) covering tops, bottoms, and dresses. To facilitate multi-reference research, we introduce DressCode-MR, built upon DressCode. As illustrated in Figure 3, it contains 28,179 samples (25,779 train / 2,400 test), each pairing a person with a complete outfit from up to five categories: tops, bottoms, dresses, shoes, and bags. We constructed this dataset by training five expert restoration models (based on CatVTON (Chong et al., 2024) and FLUX (Labs, 2024)) using VITON-HD, DressCode, and a small set of internet-sourced shoe and bag pairs. These models were used to recover the canonical images for items worn in DressCode, and the final high-quality samples were selected through human feedback.

### 4.2 IMPLEMENTATION DETAILS

We train our single-reference try-on model based on the pretrained StableDiffusion v1.5 (Rombach et al., 2021) inpainting on the DressCode (Morelli et al., 2022) and VITON-HD (Choi et al., 2021) datasets for 64,000 steps with a batch size of 32 and a resolution of 1024×768. This version is used for all single-reference quantitative evaluations. Building upon the single-reference model, we fine-tune it on our proposed DressCode-MR dataset for 16,000 steps with the same resolution and batch size. We utilized the AdamW (Loshchilov & Hutter, 2019) optimizer with a constant learning rate of $1 \times 10^{-5}$ for both training stages. To enable classifier-free guidance, 20% of the reference

images were randomly dropped during the training. All experiments were conducted on 8 NVIDIA H100 GPUs.

## 4.3 METRICS

We evaluate our model's performance on two fronts: image fidelity and computational efficiency.

**Image Fidelity.** We use two settings. In the paired setting, where ground-truth images are available, we measure similarity using the Structural Similarity Index (SSIM) (Wang et al., 2004), Learned Perceptual Image Patch Similarity (LPIPS) (Zhang et al., 2018), Fréchet Inception Distance (FID) (Seitzer, 2020), and Kernel Inception Distance (KID) (Bińkowski et al., 2021). In the unpaired setting, we assess overall realism and diversity by comparing the distribution of our generated samples to that of real images using FID and KID.

**Computational Efficiency.** We report the total parameters, inference latency, and peak memory usage. These metrics are benchmarked by averaging 100 runs on a single NVIDIA H100 GPU, with each run configured for 20 denoising steps and with classifier-free guidance (CFG) (Ho & Salimans, 2022) enabled.

## 4.4 QUANTITATIVE COMPARISON

**Single-Reference Virtual Try-On.** We conducted a quantitative comparison against current state-of-the-art virtual try-on methods (Guo et al., 2025; Kim et al., 2024; Jiang et al., 2024; Choi et al., 2024; Chong et al., 2024; Xu et al., 2024) on VITON-HD (Choi et al., 2021) and DressCode (Morelli et al., 2022) datasets. As shown in Table 3, FastFit achieves competitive results across both datasets under paired and unpaired settings, demonstrating its superior capability in generating high-quality images. Table 1 highlights the efficiency of FastFit, which achieves an average $3.5\times$ speedup over comparable methods while remaining competitive in terms of parameters and memory usage.

Table 1: Quantitative comparison of model efficiency. Best and second-best results are in **bold** and underlined, respectively. All methods are evaluated on a single H100 GPU with 20 steps. Red indicates newly added data.

| Method | Params(M)↓ | Time(s)↓ | Memory(M)↓ |
|---|---|---|---|
| OmniTry Feng et al. (2025) | 17043.63 | 51.16 | 36276 |
| Any2AnyTryon (Guo et al., 2025) | 16786.78 | 12.19 | 35218 |
| PromptDresser (Kim et al., 2024) | 6011.03 | 4.29 | 17364 |
| ITA-MDT (Hong et al., 2025) | 1891.90 | 3.44 | 28910 |
| Leffa (Zhou et al., 2024) | 1802.72 | 3.32 | 7996 |
| IDM-VTON (Choi et al., 2024) | 7086.91 | 2.76 | 19072 |
| FitDiT (Jiang et al., 2024) | 5870.80 | 2.00 | 15992 |
| OOTDiffusion (Xu et al., 2024) | 2229.73 | 1.93 | 10154 |
| IMAGDressing (Shen et al., 2024) | 2973.81 | 2.68 | 9240 |
| CatVTON (Chong et al., 2024) | **899.06** | 2.10 | **5500** |
| **FastFit** | 904.86 | **1.16** | 6944 |

**Multi-Reference Virtual Try-On.** Table 2 shows our multi-reference try-on results. In the absence of methods designed for simultaneous multi-reference generation, we adapt strong baselines from subject-driven generation (Ye et al., 2023; Yang et al., 2022; Chen et al., 2023; 2024) and multi-category try-on (Jiang

Table 2: Quantitative comparison on DressCode-MR for multi-reference try-on. Best and second-best results are in **bold** and underlined, respectively. Red indicates newly added data.

| Method | Time(s)↓ | Paired | | | | | Unpair | |
|---|---|---|---|---|---|---|---|---|
| | | FID↓ | KID↓ | SSIM↑ | LPIPS↓ | DISTS↓ | FID↓ | KID↓ |
| AnyDoor (Chen et al., 2023) | 12.08 | 37.138 | 22.571 | 0.768 | 0.235 | 0.229 | 44.068 | 23.958 |
| OmniTry (Feng et al., 2025) | 87.32 | 33.041 | 17.870 | 0.781 | 0.198 | 0.195 | 37.864 | 21.345 |
| PBE (Yang et al., 2022) | 5.22 | 28.296 | 16.092 | 0.796 | 0.215 | 0.225 | 31.135 | 17.887 |
| MimicBrush (Chen et al., 2024) | 6.62 | 21.074 | 9.858 | 0.800 | 0.173 | 0.206 | 22.111 | 9.992 |
| Part2Whole (Huang et al., 2024b) | 5.73 | 20.313 | 8.200 | 0.807 | 0.187 | 0.179 | 24.564 | 10.581 |
| CatVTON (Chong et al., 2024) | 8.94 | 16.131 | 6.980 | 0.856 | 0.106 | 0.147 | 18.339 | 7.458 |
| IP-Adapter (Ye et al., 2023) | 5.62 | 14.459 | 4.144 | **0.861** | **0.089** | 0.143 | 24.139 | 10.783 |
| FitDiT (Jiang et al., 2024) | 3.38 | 14.722 | 5.471 | 0.850 | 0.122 | 0.162 | 15.956 | 5.645 |
| **FastFit** | **1.90** | **9.311** | **1.512** | 0.859 | **0.079** | **0.117** | **12.060** | **2.124** |

et al., 2024; Chong et al., 2024; Huang et al., 2024b) via sequential single-reference inference. FastFit achieves state-of-the-art scores across quality metrics and is also the most efficient method. This demonstrates its superior ability to cohesively synthesize multiple references with high fidelity.

## 4.5 QUALITATIVE COMPARISON

**Single-Reference Virtual Try-On.** Figure 7 shows the qualitative comparison for the single-reference try-on task. On the VITON-HD (Choi et al., 2021) dataset, our method excels at preserving fine-grained details, such as the text "REBEL" on T-shirts, where other methods often produce blurred results. FastFit also realistically renders challenging materials, like the sheer polka-dot top.

Table 3: Quantitative comparison for single-reference virtual try-on on the VITON-HD (Choi et al., 2021) and DressCode (Morelli et al., 2022) datasets. All metrics are rounded to four decimal places. Best and second-best results in each column are in **bold** and underlined, respectively. Red indicates newly added data.

| Method | VITON-HD | | | | | | | DressCode | | | | | | |
|---|---|---|---|---|---|---|---|---|---|---|---|---|---|---|
| | Paired | | | | | Unpaired | | Paired | | | | | Unpaired | |
| | FID↓ | KID↓ | SSIM↑ | LPIPS↓ | DISTS↓ | FID↓ | KID↓ | FID↓ | KID↓ | SSIM↑ | LPIPS↓ | DISTS↓ | FID↓ | KID↓ |
| OmniTry (Feng et al., 2025) | 15.2550 | 6.1404 | 0.7635 | 0.2108 | 0.1946 | 15.1601 | 5.8627 | 4.1814 | 0.8845 | 0.8805 | 0.0675 | 0.0862 | 6.4109 | 1.6295 |
| Any2AnyTryon (Guo et al., 2025) | 9.9811 | 3.4964 | 0.8522 | 0.1173 | 0.0910 | 11.1953 | 2.8055 | 5.1107 | 1.2650 | 0.8966 | 0.0589 | 0.1871 | 6.7088 | 1.5797 |
| PromptDresser (Kim et al., 2024) | 5.9344 | 0.5498 | 0.8460 | 0.0902 | 0.1246 | 8.8846 | 0.9090 | 9.5625 | 4.7948 | 0.8578 | 0.1039 | 0.0919 | 10.6181 | 4.9775 |
| ITA-MDT (Hong et al., 2025) | 16.0473 | 12.0219 | 0.8477 | 0.1525 | 0.1566 | 16.4528 | 10.5733 | 11.6633 | 8.0956 | 0.8799 | 0.1255 | 0.1753 | 13.6714 | 9.1708 |
| Leffa (Zhou et al., 2024) | 5.6673 | 0.6922 | 0.8570 | 0.0762 | 0.1151 | 10.4455 | 2.6397 | 7.1932 | 2.1135 | 0.8612 | 0.0838 | 0.0855 | 20.0985 | 13.5061 |
| IDM-VTON (Choi et al., 2024) | 6.1121 | 1.1116 | 0.8655 | 0.0743 | 0.1026 | 9.2491 | 1.2672 | 7.1808 | 3.5242 | 0.8912 | 0.0695 | 0.0905 | 9.1666 | 4.4888 |
| FitDiT (Jiang et al., 2024) | 8.1763 | 1.0785 | 0.8383 | 0.0957 | 0.0917 | 9.9789 | 1.4779 | 5.5706 | 1.9007 | 0.8992 | 0.0579 | 0.1054 | 4.8045 | 0.7120 |
| IMAGDressing (Shen et al., 2024) | 8.4526 | 1.9560 | 0.8207 | 0.1292 | 0.1268 | 12.8017 | 3.9151 | 10.9586 | 5.6359 | 0.8526 | 0.1274 | 0.1665 | 15.1638 | 9.0085 |
| OOTDiffusion (Xu et al., 2024) | 5.7620 | **0.2665** | 0.8434 | **0.0724** | 0.1067 | **9.0820** | **0.7020** | 6.9754 | 2.0141 | 0.8728 | 0.0772 | **0.0833** | 8.1209 | 2.8861 |
| CatVTON (Chong et al., 2024) | 6.7382 | 1.3201 | 0.8814 | 0.0877 | 0.0916 | 10.5517 | 2.2724 | 3.7101 | 1.0104 | **0.9092** | 0.0619 | 0.0950 | 5.8715 | 1.6056 |
| **FastFit** | **5.6294** | 0.5046 | **0.8851** | 0.0778 | **0.0822** | 8.6288 | 0.6654 | **2.8361** | **0.3902** | 0.9065 | **0.0571** | 0.0850 | **4.3974** | **0.5525** |

Table 4: Ablation study of the key components in our model on DressCode (Morelli et al., 2022) dataset. The best and second-best results are demonstrated in **bold** and underlined, respectively.

| Variants | Params(M)↓ | Time (s)↓ | Memory (M)↓ | Paired | | | | Unpaired | |
|---|---|---|---|---|---|---|---|---|---|
| | | | | FID↓ | KID↓ | SSIM↑ | LPIPS↓ | FID↓ | KID↓ |
| w/o KV Cache | 904.86 | 1.92 | **6944** | 2.8585 | 0.3737 | **0.9057** | 0.0588 | 4.4206 | 0.5903 |
| w/ Full Attention | 904.86 | 2.17 | **6944** | 3.1847 | 0.5426 | 0.9056 | 0.0606 | 4.6221 | 0.6533 |
| w/o Class Embed | **904.85** | **1.16** | **6944** | 2.9146 | 0.4000 | 0.9056 | 0.0591 | 4.4624 | 0.5929 |
| w/ ReferenceNet | 1729.92 | **1.16** | 8770 | **2.8474** | **0.3577** | 0.9054 | 0.0588 | 4.4365 | **0.5741** |
| **FastFit** | 904.86 | **1.16** | **6944** | 2.8585 | 0.3737 | **0.9057** | 0.0588 | 4.4206 | 0.5903 |

On the DressCode (Morelli et al., 2022) dataset, our model accurately captures the correct shape and style of complex garments like the high-slit dress.

**Multi-Reference Virtual Try-On.** We further evaluate FastFit on the more challenging multi-reference virtual try-on task, with results presented in Figure 6. The comparison clearly demonstrates our model's superior capability. FastFit successfully synthesizes a coherent and realistic final image by seamlessly combining multiple reference items. In contrast, most existing methods, such as AnyDoor (Chen et al., 2023) and PBE (Yang et al., 2022), often fail to properly compose the different garments or produce significant artifacts. Our method, however, maintains the identity and details of each piece of clothing, resulting in a natural and believable complete outfit.

## 4.6 ABLATION STUDIES

The results in Table 4 validate our key design choices. Firstly, the Reference KV Cache is crucial for efficiency; disabling it increases inference time from 1.16s to 1.92s. It is important to clarify that this $\sim$1.66$\times$ ($\sim$40% saving) speedup is attributed solely to the caching mechanism. When combined with our holistic architectural lightweighting, FastFit achieves an aggregate $\sim$3.5$\times$ average inference speedup compared to standard diffusion-based baselines in Table 1. Crucially, this acceleration comes with no loss in generation quality, as the performance metrics are identical. Secondly, our parameter-sharing strategy is highly effective. Introducing a separate ReferenceNet nearly doubles the parameters (904.86M $\rightarrow$ 1729.9M) and increases memory usage, but yields no corresponding performance improvement. Furthermore, replacing Semi-Attention with Full Attention is detrimental, as it not only slows inference to 2.17s but also degrades generation quality (e.g., FID increases to 3.1847). We hypothesize this is because full interaction disrupts the consistency of reference features. Finally, removing the Class Embedding causes a slight performance drop, and its effectiveness in guiding region-specific attention is presented in Section 5. All ablation experiments follow the settings described in Section 4.2, trained for 32K steps, and are evaluated on the DressCode (Morelli et al., 2022) dataset.

## 4.7 FAILURE CASE ANALYSIS

As visualized in Figure 8, we analyze specific limitations when handling Out-of-Distribution (OOD) scenarios:

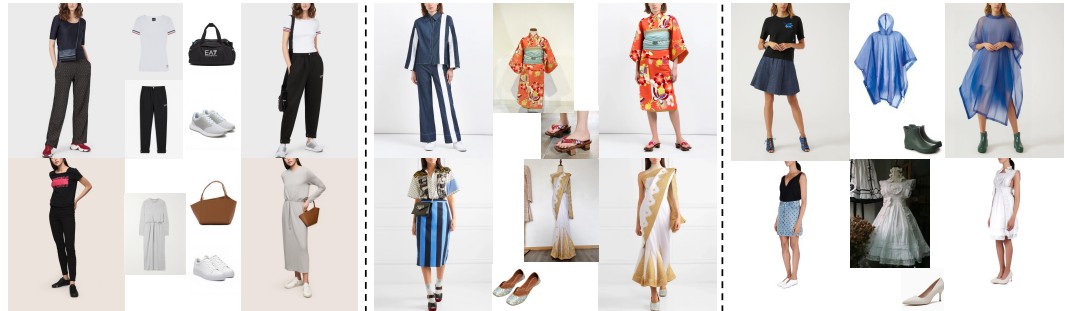

Figure 8: Failure cases. (Left) Physical Ambiguity: Accessories (e.g., bags) may lack physical grounding when the pose offers no clear interaction cues. (Middle) Structural Bias: Niche styles (e.g., Kimonos) are occasionally mapped to standard Western topologies due to dataset bias. (Right) Material Loss: Complex textures (e.g., transparent plastic) may lose high-frequency details due to their sparsity in training data.

**Physical Interaction (Left Col.):** Without explicit 3D collision modeling, the model may generate accessories (e.g., bags) with unnatural placement when the target pose lacks clear interaction cues.

**Structural Bias (Middle Col.):** For niche styles like Kimonos or Sarees, the model occasionally overfits to standard Western clothing topologies found in training data, resulting in inaccurate sleeve shapes or draping.

**Material Fidelity (Right Col.):** Challenging materials such as transparent plastic or intricate lace may exhibit texture smoothing or detail loss, primarily due to their sparsity in public datasets like VITON-HD.

## 5 CONCLUSION

In this paper, we proposed FastFit, a high-speed multi-reference virtual try-on framework designed to break the critical trade-offs between versatility, efficiency, and quality in existing technologies. Through an innovative Cacheable UNet, which combines a Class Embedding and a Semi-Attention mechanism, we decoupled reference feature encoding from the denoising process. This design enables a Reference KV Cache that allows reference features to be computed once and reused losslessly across all steps, fundamentally eliminating the computational redundancy that plagues current methods. Experimental results show that FastFit achieves a significant efficiency advantage—an average $3.5\times$ speedup over comparable methods—without sacrificing generation quality. For the first time, it enables coherent, synergistic try-on for up to 5 key categories: tops, bottoms, dresses, shoes, and bags. Furthermore, the DressCode-MR dataset we constructed provides a valuable foundation for future research in complex outfit generation. In summary, FastFit represents a promising advance towards a more realistic, efficient, and diverse virtual try-on experience, significantly lowering the barriers for its widespread application in e-commerce and intelligent outfit visualization.

**Limitations and Future Work.** Despite the model's strong performance, several areas present opportunities for future exploration. To further enhance realism, the modeling of complex physical interactions and layering among garments could be improved. Expanding the DressCode-MR dataset with such complex interaction pairs would be a valuable direction. Another important research path is improving generalization to underrepresented apparel, such as styles with unique topologies or challenging materials. Finally, while our framework significantly accelerates inference, a gap remains toward achieving real-time interaction. Exploring techniques such as guidance and step distillation, combined with more advanced caching mechanisms, offers a promising path to bridge this gap and enable applications like interactive real-time outfit visualization.

## ETHICS STATEMENT

We recognize the ethical implications associated with generative models and datasets involving human imagery. Our work is built upon publicly available datasets, specifically VITON-HD and Dress-Code, and we adhere to the ethical guidelines established by their original authors. For our newly constructed DressCode-MR dataset, all source images are derived from the publicly available and curated DressCode dataset, thus respecting the original data subjects' privacy and usage agreements. Our framework, FastFit, is intended for beneficial applications in virtual e-commerce and interactive outfit visualization, aiming to enhance the user shopping experience. We strongly oppose any potential misuse of this technology, such as the generation of misleading or unauthorized images. We acknowledge that image-based person datasets may contain inherent biases related to demographic representation (e.g., body shape, skin tone, clothing style). While the scope of this work is methodological, we recognize the importance of fairness and diversity and commit to actively addressing and mitigating these biases in future research and dataset construction efforts. All methodologies and results are presented with full transparency to maintain research integrity.

## REPRODUCIBILITY STATEMENT

To ensure reproducibility, we provide this anonymous link, including our source code, model weights and DressCode-MR dataset. The methods described in Section 3 offer a clear technical blueprint for implementing our architecture. Detailed implementation specifics, such as training configurations, optimizer settings, and hyperparameters, are provided in Section 4.2. Using the information in these sections and the code provided, we believe that an independent researcher can reproduce our experimental results presented in Section 4.4 and Section 4.6.

### LLM USAGE

We acknowledge the use of LLMs as a writing assistant. The LLMs are utilized solely to assist with proofreading, grammatical correction, and minor stylistic refinements of the manuscript.

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

APPENDIX

A. QUANTITATIVE COMPARISON ACROSS GARMENT TYPES

For a more fine-grained analysis, Table 5 presents a quantitative comparison on the DressCode (Morelli et al., 2022) dataset, with results broken down by clothing category. The results highlight the robust and superior performance of our method across all tested categories, including upper, lower, and dresses. FastFit consistently achieves either the best or second-best scores in the vast majority of key metrics, demonstrating its strong and stable performance regardless of the garment type. This showcases the model's excellent generalization capability for different clothing styles.

Table 5: Quantitative comparison on the DressCode dataset, with results broken down by category (Upper, Lower, and Dresses). The best results are marked in **bold** and the second-best are underlined. ↓ indicates lower is better, while ↑ indicates higher is better.

| Methods | Upper | | | | Lower | | | | Dresses | | | |
|---|---|---|---|---|---|---|---|---|---|---|---|---|
| | FID↓ | KID↓ | SSIM↑ | LPIPS↓ | FID↓ | KID↓ | SSIM↑ | LPIPS↓ | FID↓ | KID↓ | SSIM↑ | LPIPS↓ |
| Any2AnyTryon | 10.4741 | 1.7130 | 0.9206 | 0.0476 | 13.1152 | 2.8336 | 0.8896 | 0.0655 | 9.1124 | 1.6539 | 0.8796 | 0.0636 |
| PromptDresser | 9.2447 | 0.7174 | 0.9044 | 0.0678 | 32.9093 | 17.9749 | 0.8327 | 0.1352 | 16.8179 | 6.8932 | 0.8363 | 0.1087 |
| Leffa | 11.2549 | 2.0947 | 0.8908 | 0.0578 | 19.6834 | 5.8985 | 0.8594 | 0.0908 | 13.3859 | 2.3701 | 0.8335 | 0.1029 |
| IDM-VTON | 11.2283 | 3.3860 | 0.9174 | 0.0547 | 11.7878 | 3.4218 | 0.8978 | 0.0655 | 17.5135 | 8.3389 | 0.8585 | 0.0882 |
| OOTDiffusion | 9.5945 | 1.1055 | 0.9040 | 0.0528 | 19.6615 | 6.1217 | 0.8751 | 0.0827 | 14.8496 | 4.4567 | 0.8393 | 0.0963 |
| CatVTON | 7.8465 | 1.0851 | **0.9360** | 0.0504 | 8.6135 | 1.6574 | **0.9236** | 0.0562 | 8.9453 | 1.0575 | 0.8669 | 0.0791 |
| FitDiT | 8.0876 | 0.5789 | 0.9241 | **0.0417** | 24.5079 | 11.7225 | 0.8944 | 0.0758 | **7.2253** | 0.4768 | **0.8789** | **0.0562** |
| **FastFit** | **6.8354** | **0.2453** | 0.9318 | 0.0485 | **7.1311** | **0.7981** | 0.9207 | **0.0511** | 7.5890 | **0.2446** | 0.8671 | 0.0720 |

B. MORE VISUAL COMPARISONS

B.1. SINGLE-REFERENCE VIRTUAL TRY-ON

In the single-reference virtual try-on task, our method demonstrates robust performance across both the VITON-HD (Choi et al., 2021) and DressCode (Morelli et al., 2022) datasets. As illustrated in Figure 11, FastFit excels at preserving high-frequency details on the garments, such as intricate patterns and text logos. Furthermore, our model accurately renders the correct shape and length for various types of clothing. The final results show that the garments are naturally fused with the person's body, effectively handling challenging poses and occlusions.

B.2. MULTI-REFERENCE VIRTUAL TRY-ON

For the more challenging multi-reference task, FastFit exhibits a significant advantage over competing methods. Figure 12 showcases our model's unique ability to seamlessly combine multiple, distinct reference items into a single, coherent outfit. Notably, even during this complex composition process, FastFit faithfully preserves the fine-grained details and logos of each individual item (e.g., "SPAS", "CHIUS"). This capability to generate complete and detailed ensembles in complex scenarios highlights its superiority where other methods often struggle.

B.3. QUALITATIVE COMPARISONS WITH CLOSED-SOURCE MODELS

To more comprehensively evaluate our model's performance, we conducted qualitative comparisons with mainstream closed-source models, including Nano Banana, GPT-4o, and FLUX.1 Kontext (Black Forest Labs et al., 2025) [Pro], on the multi-reference virtual try-on task. Due to their input limitations, we stitched the source person and multiple reference images into a single composite image. We then used the prompt, "Dress the person on the left with the garments and accessories on the right," to obtain the results. Furthermore, given the non-deterministic nature of their outputs, we generated four results for each example and selected the most plausible one for display. As seen in Figure 9, our method is better at preserving the original person's pose and background environment, and more consistently integrates details from the various reference images.

C. VISUAL ANALYSIS OF THE EFFECTIVENESS OF CLASS EMBEDDINGS

To visually validate the effectiveness of the Reference Class Embedding as a key control mechanism in our model, we conducted an additional ablation study. As shown in Figure 10, the experiment is designed to isolate the influence of the class embedding. For each example row, we provide the model with the exact same source person and reference image. The only variable changed across the columns is the specific class embedding provided (e.g., 'Upper', 'Lower', 'Dresses', 'Shoes',

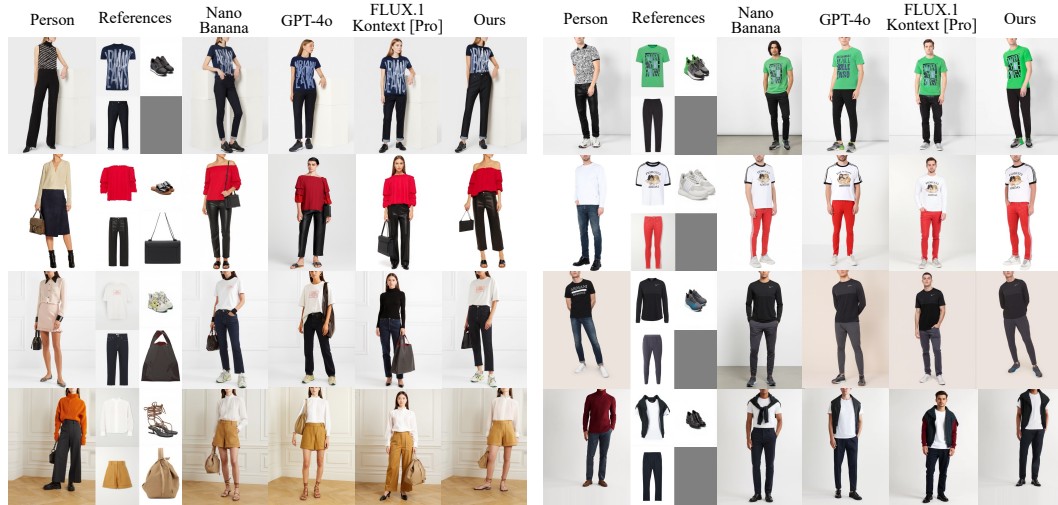

Figure 9: Qualitative comparison with mainstream closed-source models on multi-reference virtual try-on. This figure provides a direct visual comparison of our model against several top-tier closed-source models (Nano Banana, GPT-4o, FLUX.1 Kontext (Black Forest Labs et al., 2025) [Pro]) on the multi-reference virtual try-on task.

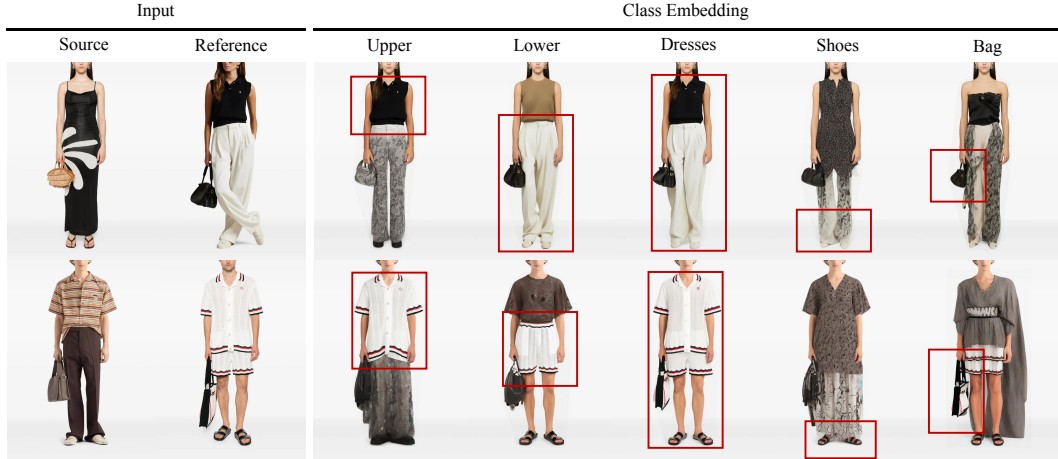

Figure 10: Demonstration of the visual effect of class embeddings. By providing different class embeddings (e.g., 'Upper', 'Lower', 'Dresses', 'Shoes', 'Bag') for the same reference image, our model can be directed to selectively transfer the corresponding item to the source person.

'Bag'). The results demonstrate that the class embedding provides fine-grained, semantic control over the try-on process. The model is able to interpret the embedding and selectively transfer the corresponding item from the reference image, even when multiple items are present. This experiment confirms that by applying a Class Embedding, the model's attention is effectively guided to the corresponding region of the reference image, which is crucial for preventing the features of different reference items from being conflated in a multi-reference scenario.

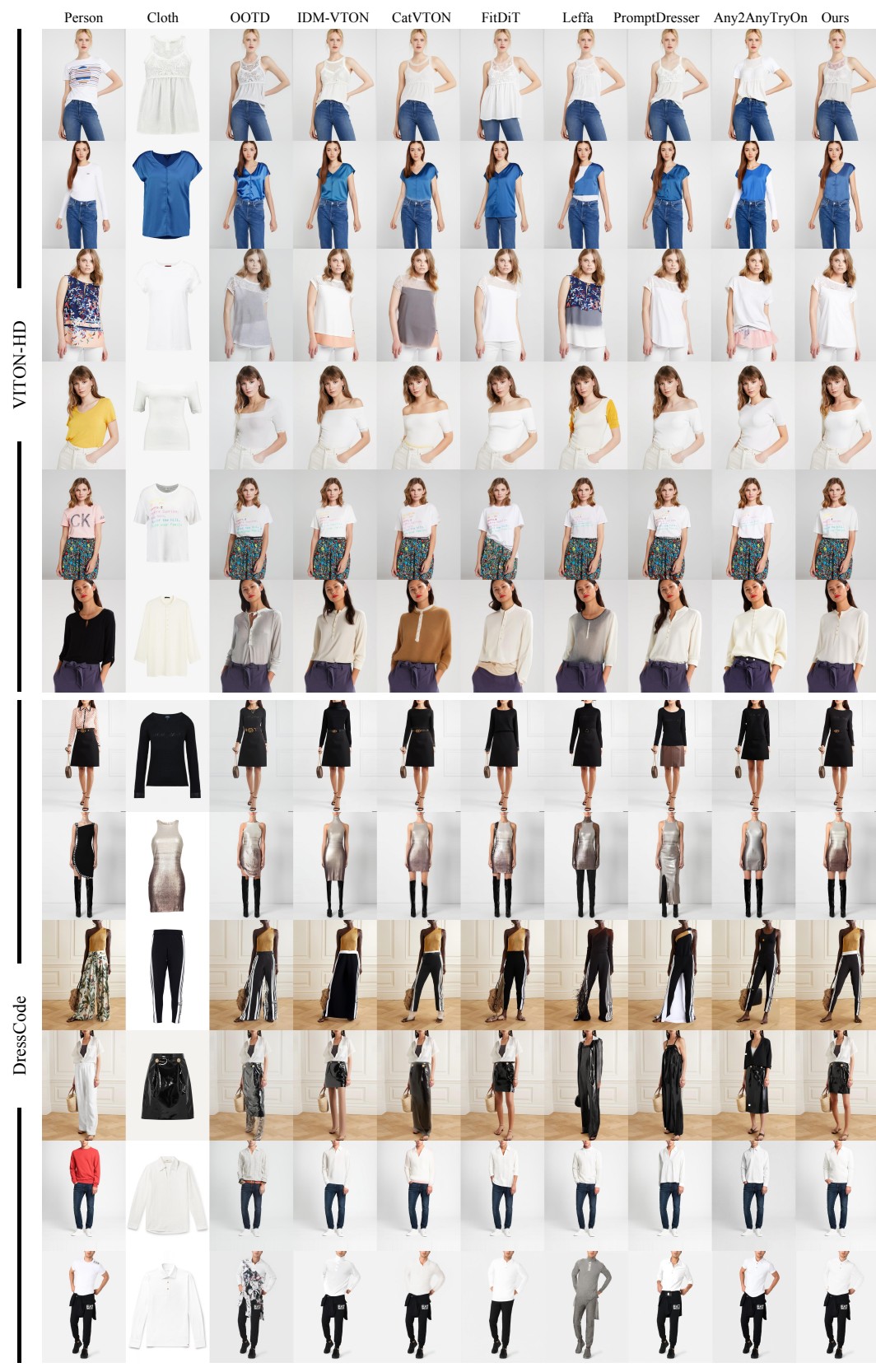

Figure 11: More visual comparisons on the VITON-HD (Choi et al., 2021) and DressCode (Morelli et al., 2022) dataset with baseline methods. Please zoom in for more details.

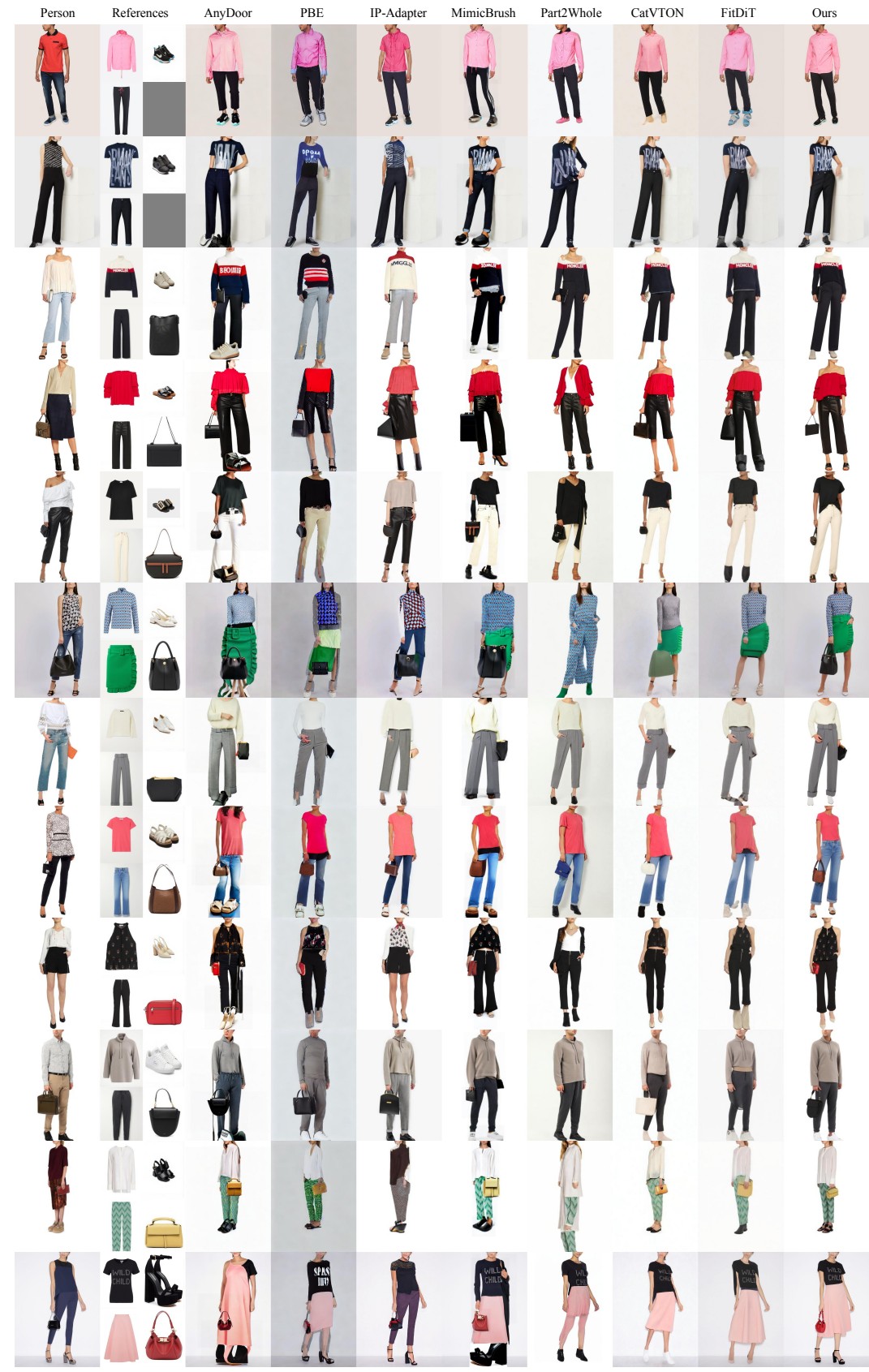

Figure 12: More visual comparisons on the DressCode-MR dataset with baseline methods. Please zoom in for more details.

