# OpenReview forum: "FastFit: Accelerating Multi-Reference Virtual Try-On via Cacheable Diffusion Models"
_ICLR.cc/2026/Conference — Submitted to ICLR 2026_

### Official Review · Reviewer_Z1Vn · 2025-10-27

**Soundness:** 3
**Presentation:** 1
**Contribution:** 2
**Rating:** 4
**Confidence:** 5

**Summary:**

This paper presents **FastFit**, a high-speed multi-reference virtual try-on framework that addresses current limitations in handling multiple outfit items and redundant feature recomputation.

By introducing a cacheable diffusion architecture with a Semi-Attention mechanism and class embeddings for reference items, the method decouples feature encoding from denoising, achieving up to **3.5× faster inference** with minimal overhead.

The authors also introduce **DressCode-MR**, a new dataset with over 28K curated image pairs, demonstrating that FastFit outperforms state-of-the-art methods in both fidelity and efficiency.

**Strengths:**

1. The paper provides extensive empirical results across multiple benchmarks (VITON-HD, DressCode, and DressCode-MR) to demonstrate the model’s performance.
2. The proposed FastFit framework effectively addresses real-world virtual try-on challenges by enabling multi-reference outfit compositions with improved efficiency.
3. The introduction of the large-scale DressCode-MR dataset enriches the field with diverse, high-quality samples for evaluating complex virtual try-on scenarios.

**Weaknesses:**

* The novelty appears limited, as the method largely reuses existing components. The proposed semi-attention with KV caching closely resembles techniques in **OminiControl2 (2025)** and **EasyControl (2025)**, while other modules (e.g., concatenation and cross-attention) are also standard design choices.
* Some recent and efficient baselines with strong detail preservation are missing from the comparison, such as **ITA-MDT (CVPR 2025)** and **IMAGDressing-v1 (arXiv:2407.12705)**.
* The paper’s clarity requires substantial improvement, as several explanations and figures are confusing or incomplete:

    (1) *Line 80*: The phrase “which avoids this redundancy” is unclear—what specific redundancy is being referred to, and how does reference-based modeling reduce it?

    (2) *Figure 4*: The relationship between sub-figures (a) and (b) is ambiguous. (a) shows seven inputs, whereas (b) only depicts four. The right part of (b) further introduces two small diagrams with inconsistent inputs (X with R1–R3 vs. X alone), making the overall data flow unclear.

    (3) Several notations (e.g., X′ in Figure 4) are unexplained. Important variables mentioned in the text should be explicitly labeled in the figure for coherence. Additionally, Figure 4(b) refers to a “Cacheable UNet Block,” but it is unclear how this integrates into the overview in (a) and how many such blocks are used. An ablation study analyzing this design would be valuable.

    (4) *Lines 268–269*: The explanation of timestep embeddings is inconsistent. The paper claims to explore timestep embeddings for enhanced control but then discusses only class embeddings. How exactly do class embeddings contribute to timestep control?

    (5) Baselines are listed without proper citations, making it difficult to trace their corresponding works. Please include references rather than assuming readers are familiar with each method by name.

    (6) The efficiency table lacks sufficient detail—specifically, the number of inference steps used for each baseline should be provided for fair comparison.

    (7) Please clarify Figure 4. Is that for training or inference?

[1] ITA-MDT: Image-Timestep-Adaptive Masked Diffusion Transformer Framework for Image-Based Virtual Try-On, CVPR 2025

[2] IMAGDressing-v1: Customizable Virtual Dressing, arXiv:2407.12705

**Questions:**

Weakness part

---

> ### Author Response · Authors · 2025-11-30
> **Response to Reviewer Z1Vn**
>
> We sincerely thank the reviewer for the rigorous assessment and detailed feedback. We have meticulously revised the paper, particularly Figure 4 and the Method section, to address every point raised. Below is our point-by-point response:
>
> ### **1. Comparison with OmniControl2/EasyControl**
>
> We clarify that while OmniControl2 and EasyControl also possess subject-driven generation capabilities, they fundamentally lack the composability for multiple subjects. In contrast, FastFit is specifically designed for Multi-Reference Composition. This enables the coherent synthesis of not just garments but also diverse accessories (e.g., bags, shoes) simultaneously within a single pass—a capability not supported by the aforementioned methods. We have revised the Introduction and Method sections to clearly articulate our motivation.
>
> ### **2.Missing Baselines (ITA-MDT, IMAGDressing-v1)**
>
> We thank the reviewer for identifying these state-of-the-art benchmarks. We have added quantitative comparisons with OmniTry, ITA-MDT, and IMAGDressing-v1 in Table 1 and Table 3. Across these comparisons, our method consistently outperforms these baselines, validating its superior generation quality and efficiency.
>
> ### **3.Response to Clarity Issues (Point-by-Point)**
>
> **(1) Line 80 "Redundancy":** We refer to the computational redundancy in In-Context Learning methods. In those approaches, reference features are re-calculated at every denoising step due to the concatenation mechanism. In contrast, reference-based modeling (like ours and ReferenceNet) extracts features only once. We have rewritten this paragraph in the Introduction to explicitly define this redundancy and improve the logical flow.
>
> **(2) & (3) Figure 4 Ambiguities & Notations:** We apologize for the confusion. We have completely redesigned Figure 4.
>  The inputs in sub-figures (a) and (b) are now strictly consistent. We added explicit equations and text descriptions for X and other variables to match the figure labels. We clarified that all U-Net blocks in the Denoising Network are converted into "Cacheable UNet Blocks" to enable full-scale reference injection.
>
> **(4) Timestep vs. Class Embeddings (Lines 268-269):**  Our method substitutes the standard timestep embedding input of the Reference U-Net with Class Embeddings.  Existing methods typically input a zero-vector for the timestep in the Reference U-Net. We found this wasteful. Instead, we inject category information (e.g., "shirt", "pants") into this slot. This "tricks" the network into processing the image with category-specific attention, significantly enhancing the distinctness of features in multi-garment scenarios (as shown in Ablation Fig. 9). We have corrected the text to accurately describe this substitution strategy.
>
> **(5) Citation of Baselines:** We have added formal citations for all baselines in the tables and text to ensure traceability.
>
> **(6) Efficiency Table Details:** We wish to clarify that these experimental details (20 inference steps, CFG enabled, averaged over 100 runs on an NVIDIA H100) were **already reported in Section 4.3 (Metrics) of the original submission**. To prevent future oversight and ensure immediate clarity, we have explicitly repeated these settings in the caption of Table 1 in the revised manuscript.
>
> **(7) Figure 4 Context:** Figure 4 illustrates the Inference Pipeline. We have updated the caption to explicitly state this.
>
>
> We believe these substantial revisions to the presentation and the inclusion of new baselines fully address the reviewer's concerns.
> We sincerely thank the reviewer again for the constructive feedback, which has significantly helped us improve the clarity and quality of our presentation.

---

### Official Review · Reviewer_mwM1 · 2025-10-27

**Soundness:** 2
**Presentation:** 3
**Contribution:** 2
**Rating:** 4
**Confidence:** 5

**Summary:**

This paper proposes FastFit, a diffusion-based framework designed to address two key challenges in virtual try-on (VTON) technology: the inability of existing methods to simultaneously compose multiple garments/accessories (e.g., tops, shoes, bags) and the computational redundancy caused by re-computing reference garment features at every denoising step. At the core of FastFit is a Cacheable UNet architecture, which decouples reference feature encoding from the iterative denoising process through two innovations: Reference Class Embedding and Semi-Attention Mechanism. This design enables a Reference KV Cache, where reference features are pre-computed once and reused across all denoising steps—delivering an average 3.5× speedup over comparable methods. To support multi-reference VTON research, the authors also introduce DressCode-MR, a large-scale dataset with 28,179 samples covering five categories (tops, bottoms, dresses, shoes, bags), curated via expert models (e.g., CatVTON) and human feedback.

**Strengths:**

- Most existing VTON methods (e.g., CatVTON, Chong et al. 2024; FitDiT, Jiang et al. 2024) only support single-garment try-on, requiring sequential inference for multi-item outfits (introducing error accumulation and latency). FastFit is among the first to enable simultaneous composition of garments and accessories (including shoes and bags), a critical capability for realistic outfit visualization. Qualitative results confirm it preserves fine details (e.g., text logos on T-shirts, sheer fabrics) across multiple references, outperforming baselines that produce blurred or misaligned components .
- Prior methods face a trade-off: ReferenceNet-based approaches (e.g., Xu et al. 2024) avoid redundancy but add massive parameter overhead (e.g., 1.7B params vs. FastFit’s 904M), while in-context learning methods (e.g., Huang et al. 2024a) re-compute features at each step (2–6× slower). FastFit’s decoupled design breaks this trade-off: ablations show disabling the Reference KV Cache increases latency by 1.66× (1.16s → 1.92s) with no quality loss, confirming its efficiency gain .
- Public VTON datasets (VITON-HD, DressCode) lack multi-reference pairs. DressCode-MR, built via expert model-based canonical image restoration and human feedback, provides the first large-scale benchmark for multi-reference VTON. This enables standardized evaluation of complex outfit composition, a previously under-explored direction

**Weaknesses:**

- The paper’s focus on multi-reference VTON overlaps with OmniTry (a previously published work on "try-on anything" that supports diverse garment/accessory categories). FastFit does not explicitly compare with OmniTry or demonstrate unique advantages in category coverage, generality, or composition flexibility—undermining its claim of advancing multi-reference VTON. This overlap reduces the work’s incremental contribution.
- The paper fails to cite key recent VTON research, such as VTON-HandFit (CVPR 2025), which addresses hand-garment interaction (a critical realism cue for accessories like bags or gloves). This omission limits the contextualization of FastFit’s contributions: VTON-HandFit’s techniques for handling fine-grained garment-object interactions could have informed FastFit’s multi-reference design, and ignoring it weakens the paper’s academic rigor.
- Training requires 8 NVIDIA H100 GPUs, and inference depends on high-end hardware (H100). No evaluation on consumer GPUs (e.g., RTX 4090) or edge devices limits real-world deployment—contrast with lightweight baselines like Leffa (1.8B params) that run on edge hardware , .
- FastFit cannot capture physical interactions (e.g., a tucked shirt, a jacket over a top) or fit variations (loose/tight clothing). Unlike OutfitAnyone (Sun et al. 2024) (which uses 3D modeling for layering), FastFit’s 2D diffusion approach produces flat, unrealistic composites—evident in qualitative results where accessories like bags lack depth relative to the body , .
- Evaluations focus on common categories (tops, dresses) but omit niche styles (e.g., sarees, kimonos) or challenging materials (e.g., leather, lace). The paper acknowledges this gap but provides no failure case analysis, limiting confidence in its applicability to diverse real-world scenarios .

**Questions:**

- How does FastFit differ from OmniTry in terms of category coverage, composition flexibility, and realism? What unique capabilities does FastFit offer that OmniTry lacks?
- Could integrating 3D garment priors (e.g., 3DMM) improve FastFit’s ability to model layering and physical interactions? Would this compromise its efficiency advantage?

---

> ### Author Response · Authors · 2025-11-30
> **Response to Reviewer mwM1**
>
> We thank the reviewer for the constructive feedback. We address the reviewer's concerns point-by-point below:
>
> ## **Response to Weaknesses**
>
> ### **1. Overlap with OmniTry [1]**
>
> We respectfully clarify that OmniTry [1]  is a **concurrent work (released on ArXiv in August 2024, around our submission time)**. Despite this, we have added a comprehensive comparison in the revision to demonstrate FastFit's superiority: OmniTry generates garments sequentially (**one-by-one inpainting, over 50 seconds for one**), which limits global consistency and time cost. In contrast, FastFit employs a joint multi-reference architecture, synthesizing multiple garments at once with **faster speed (within 2 seconds)**. As shown in **Tables 1, 2, and 3** (highlighted in *red*), FastFit surpasses OmniTry in both single/multi-reference quantitative metrics and efficiency metrics.
>
> ### **2. Comparison with VTON-HandFit  [2]**
>
> We appreciate the pointer and have added the citation and discussion of VTON-HandFit [2]. Regarding comparison: We actively attempted to compare with VTON-HandFit. However, while their code is public, the model weights are not released. We have emailed the authors to request access. Due to this unavailability, a fair quantitative comparison is currently infeasible. We commit to adding this comparison in the final version after the weights become available. Besides, we have expanded our experiments by adding quantitative comparisons with **ITA-MDT [4] , and IMAGDressing-v1 [5] in Tables 1 and 3**. These results further validate FastFit’s superiority against a broader range of SOTA methods.
>
> ### **3. Hardware Requirements & Comparison with Leffa [3]**
>
> There is a significant misunderstanding regarding our inference costs. While we used H100s for training, FastFit is highly efficient during inference. We respectfully correct the premise: FastFit outperforms Leffa in VRAM usage, Inference Time, and Parameters. As shown in **Table 1**, FastFit requires only **~7GB VRAM** (vs. Leffa's **~8GB**), making it fully deployable on consumer GPUs like the **RTX 3090/4090**, contrary to the concern.
>
> ### **4. Physical Interactions & OutfitAnyone [6]**
>
> We respectfully wish to clarify a technical premise. To the best of our knowledge, **OutfitAnyone [6] is also a 2D diffusion-based framework and does not explicitly utilize 3D modeling for layering**. Thus, FastFit and OutfitAnyone share the same 2D paradigm. For visual effect, we invite the reviewer to **re-examine Figures 6, 7, 9, 11, 12**. In side-by-side comparisons, FastFit demonstrates SOTA-level detail retention and realistic results among 2D methods. Complex physical interactions (e.g., tuck-ins) are a shared challenge for all pure 2D diffusion models (including OutfitAnyone). We explicitly discuss this in the "Limitations" section as a key direction for future research, rather than a specific deficiency of FastFit relative to its peers.
>
> ### **5. Niche Styles & Failure Analysis**
>
> Our work focuses on unified multi-reference composition and efficiency using standard benchmarks (VITON-HD, DressCode). The lack of niche styles (e.g., sarees) stems from inherent biases in these public datasets, not the FastFit architecture itself. We agree that failure analysis is vital. We have added a **Failure Case Analysis in Section 4.7**, evaluating out-of-distribution examples (e.g., Structural Bias & Material Loss) to provide a transparent view of the model's boundaries.
>
> ## **Response to Questions**
>
> - **Q1 (Difference from OmniTry):** FastFit differs fundamentally in architecture (**Joint generation** vs. OmniTry's **Sequential** generation), leading to better global coherence and significantly higher efficiency (see Response to W1).
>
> - **Q2 (3D Priors):** Yes, integrating 3D priors (e.g., 3DMM, depth maps) could improve layering physics. However, this often introduces significant computational overhead (preprocessing/rendering). Our current design prioritizes **real-time applicability**, but we agree that lightweight 3D guidance is a promising future direction.
>
>
> We hope that our clarifications regarding these misunderstandings and factual errors will assist the reviewer in re-evaluating our work more accurately.

---

> > ### Author Response · Authors · 2025-11-30
> > **References**
> >
> > ### **References:**
> >
> > [1] Feng, Y., Zhang, L., Cao, H., Chen, Y., Feng, X., Cao, J., Wu, Y., & Wang, B. (2025). OmniTry: Virtual Try-On Anything without Masks (No. arXiv:2508.13632). arXiv. https://doi.org/10.48550/arXiv.2508.13632
> >
> > [2] Liang, Y., Hu, X., Jiang, B., Luo, D., WU, K., Han, W., Jin, T., & Wang, C. (2024). VTON-HandFit: Virtual Try-on for Arbitrary Hand Pose Guided by Hand Priors Embedding (No. arXiv:2408.12340). arXiv. https://doi.org/10.48550/arXiv.2408.12340
> >
> > [3] Zhou, Z., Liu, S., Han, X., Liu, H., Ng, K. W., Xie, T., Cong, Y., Li, H., Xu, M., Pérez-Rúa, J.-M., Patel, A., Xiang, T., Shi, M., & He, S. (2024). Learning Flow Fields in Attention for Controllable Person Image Generation (No. arXiv:2412.08486). arXiv. https://doi.org/10.48550/arXiv.2412.08486
> >
> > [4] Hong, J. W., Ton, T., Pham, T. X., Koo, G., Yoon, S., & Yoo, C. D. (2025). ITA-MDT: Image-Timestep-Adaptive Masked Diffusion Transformer Framework for Image-Based Virtual Try-On (No. arXiv:2503.20418). arXiv. https://doi.org/10.48550/arXiv.2503.20418
> >
> > [5] Shen, F., Jiang, X., He, X., Ye, H., Wang, C., Du, X., Li, Z., & Tang, J. (2024). IMAGDressing-v1: Customizable Virtual Dressing (No. arXiv:2407.12705). arXiv. http://arxiv.org/abs/2407.12705
> >
> > [6] Sun, K., Cao, J., Wang, Q., Tian, L., Zhang, X., Zhuo, L., Zhang, B., Bo, L., Zhou, W., Zhang, W., & Gao, D. (2024). OutfitAnyone: Ultra-high Quality Virtual Try-On for Any Clothing and Any Person (No. arXiv:2407.16224). arXiv. http://arxiv.org/abs/2407.16224

---

### Official Review · Reviewer_YopD · 2025-10-31

**Soundness:** 3
**Presentation:** 3
**Contribution:** 2
**Rating:** 6
**Confidence:** 4

**Summary:**

The authors identify two key limitations of current methods: their inability to support multi-reference outfit compositions and their significant computational inefficiency. To address these issues, they introduce a new large-scale dataset, DressCode-MR, and propose a Semi-Attention mechanism combined with cached features to accelerate the try-on process, reportedly saving approximately 40% inference time.

**Strengths:**

1. The writing is clear and the methodology is easy to understand.

2. The ablation studies thoroughly validate each component of the proposed approach—I appreciate the completeness of the ablation analysis.

3. In fact, the primary bottleneck preventing prior models from supporting multi-reference outfit composition has been the lack of suitable datasets. The most significant contribution of this paper is the introduction of the new multi-garment virtual try-on dataset, DressCode-MR.
However, the authors do not explicitly commit to open-sourcing DressCode-MR. If the authors clearly promise to fully release the DressCode-MR dataset (including both training and test sets), I would be happy to reconsider and potentially raise my rating.

**Weaknesses:**

1. The main concern is limited novelty. Multi-garment try-on has already been explored in prior works such as MMTryon [1] and AnyFit [2], which adopt similar strategies—e.g., spatially concatenating multiple garment references. In my view, using dedicated trainable branches for conditions versus sharing parameters with the denoising branch is primarily an engineering-level optimization rather than a fundamental academic distinction. One could interpret a single denoising branch handling both conditions and noise as an instance of shared trainable branches.

2. Although KV caching has not been explicitly framed as a novelty in prior try-on literature, it has already been implicitly applied in practice, which diminishes its perceived academic contribution. Moreover, the reported 40% speedup feels modest—almost trivial—for a paper whose title emphasizes acceleration. Notably, AnyFit [2] already employs KV caching or frozen condition features, yet the paper does not discuss or compare against AnyFit. I find this omission surprising and recommend that the authors add a dedicated discussion in the main body of the paper.

3. Additional evaluation metrics—such as DISTS [3]—would strengthen the experimental analysis.

4. It would also be beneficial to include visual results corresponding to the ablation studies.

Despite the above concerns, I believe the most valuable contribution of this work is the DressCode-MR dataset, which has the potential to significantly advance research in multi-reference virtual try-on and outweigh the methodological limitations.
Therefore, my final rating heavily depends on whether the authors commit to fully open-sourcing both the training and test sets of DressCode-MR.

References:

[1] MMTryon: Multi-Modal Multi-Reference Control for High-Quality Fashion Generation.

[2] AnyFit: Controllable Virtual Try-on for Any Combination of Attire Across Any Scenario.

[3] Image Quality Assessment: Unifying Structure and Texture Similarity.

**Questions:**

Please refer to the Weaknesses.

---

> ### Author Response · Authors · 2025-11-30
> **Response to Reviewer YopD**
>
> We sincerely thank the reviewer for the constructive feedback and for recognizing **the value of our DressCode-MR dataset**. And we explicitly commit to **fully open-sourcing the entire DressCode-MR dataset (both Training and Test sets), along with the complete model weights and code**.  We agree that the lack of high-quality datasets is a major bottleneck, and we aim to facilitate future research in multi-reference virtual try-on through this release.
>
> We address the reviewer's concerns point-by-point below:
>
> ### **1. Novelty & Comparison with MMTryon/AnyFit.**
>
> We respectfully clarify that MMTryon and AnyFit represent the "ReferenceNet-based" paradigm (illustrated in our Figure 2a), which typically requires training separate parameters or distinct networks for different garment types (e.g., separate encoders for upper vs. lower body). In-Context Learning (Figure 2b) methods do not require additional networks, but performing Feature Caching is difficult. **FastFit integrates the strengths of both In-Context Learning and ReferenceNet-based approaches: reducing parameter count and achieving high inference efficiency.** We have revised the Introduction and Method sections to clearly articulate this hybrid motivation. Regrettably, as neither MMTryon nor AnyFit has released their source code or pre-trained models, we are unable to provide quantitative comparisons.
>
> ### **2. KV Cache Novelty & Speedup Magnitude.**
>
> Pleas refer to Response to **Reviewer syrZ (A1)** for discussion on why our caching strategy is an architectural byproduct rather than a mere engineering trick. Regarding the Speedup Magnitude, the "40% speedup" refers specifically to the ablation study comparing FastFit with vs. without caching. It does not represent our total speedup against other state-of-the-art methods. Our efficiency gains are not solely from caching but also from architectural streamlining. By removing redundant modules found in standard pipelines (e.g., CLIP Encoders, PoseNets, and heavy Cross-Attention layers), FastFit achieves **an average inference speedup of 3.5x** compared to the baselines listed in Table 1. As mentioned in Point 1, since AnyFit is not open-source, we cannot compare directly. Instead, we have **added comparisons with OmniTry [1], ITA-MDT [2], and IMAGDressing-v1[3] in Tables 1, 2, and 3** to comprehensively demonstrate our method's superiority in both efficiency and quality.
>
> ### **3. Additional Metrics (DISTS).**
>
> We have included the DISTS metric in **Table 2 and Table 3** of the revised manuscript. All newly added results are highlighted in *red* for easier reference. FastFit achieves state-of-the-art (best or second-best) performance across three datasets under this metric, further validating the structural and textural fidelity of our model.
>
> ### **4. Visual Ablations.**
>
> Due to page limits in the main text, the visual results corresponding to the ablation studies (specifically regarding Class Embedding) have been provided in **Appendix C and Figure 10** of the original submission.
>
> We believe the full release of DressCode-MR, combined with the proven efficiency (3.5x speedup) and unified architecture of FastFit, constitutes a solid contribution to the community.
>
> ### **References:**
>
> [1] Feng, Y., Zhang, L., Cao, H., Chen, Y., Feng, X., Cao, J., Wu, Y., & Wang, B. (2025). OmniTry: Virtual Try-On Anything without Masks (No. arXiv:2508.13632). arXiv. https://doi.org/10.48550/arXiv.2508.13632
>
> [2] Hong, J. W., Ton, T., Pham, T. X., Koo, G., Yoon, S., & Yoo, C. D. (2025). ITA-MDT: Image-Timestep-Adaptive Masked Diffusion Transformer Framework for Image-Based Virtual Try-On (No. arXiv:2503.20418). arXiv. https://doi.org/10.48550/arXiv.2503.20418
>
> [3] Shen, F., Jiang, X., He, X., Ye, H., Wang, C., Du, X., Li, Z., & Tang, J. (2024). IMAGDressing-v1: Customizable Virtual Dressing (No. arXiv:2407.12705). arXiv. http://arxiv.org/abs/2407.12705

---

### Official Review · Reviewer_syrZ · 2025-11-02

**Soundness:** 3
**Presentation:** 2
**Contribution:** 2
**Rating:** 4
**Confidence:** 5

**Summary:**

This paper primarily addresses two issues: first, it improves the efficiency of ReferenceNet-style garment swapping by caching features from reference images. Second, it introduces a dataset named DressCode-MR, which includes both garments and accessories, enabling simultaneous garment and accessory swapping.

**Strengths:**

It achieves more efficient injection of reference information by caching the features of the reference image, while also providing a dataset of garments and accessories with significant academic value. The paper is clearly written and easily understandable, with extensive experiments conducted.

**Weaknesses:**

The most significant issue with this paper is that its core algorithmic contribution lies in proposing a method to cache reference image information. However, such a technique is already widely used in both the research community and industrial applications. It appears more like a trick rather than a novel algorithm, which is my primary concern.

At the same time, compared to Anyfit [1], this paper does introduce an approach for simultaneous multi-garment replacement. The experimental results strike me as somewhat puzzling: why does the proposed method, with fewer parameters, outperform ReferenceNet? Caching reference image features should not enhance model performance—it only improves inference efficiency. The parameters in ReferenceNet enable it to better adapt to the features provided by the Denoising U-Net. In fact, in practical applications, we often precompute and store ReferenceNet's features. Doesn’t this approach closely resemble the method proposed in the paper? This similarity further undermines the core contribution of this work.

[1] Anyfit: Controllable virtual try-on for any combination of attire across any scenario

**Questions:**

Please refer to Weaknesses

---

> ### Author Response · Authors · 2025-11-30
> **Response to Reviewer syrZ**
>
> We thank the reviewer for acknowledging **the academic value of our dataset DressCode-MR and the extensive experiments**.
> Below, we address the concerns about novelty and performance gains to clear up the confusion.
>
> ### **Q1: Concern that "caching reference features" is just a widely used trick, not a novel algorithm.**
>
> **A1:** We clarify that feature caching is a byproduct of our architectural efficiency. By combining Class Embeddings with Semi-Attention, we explicitly decouple reference feature from the denoising process. This design empowers the model to naturally try-on multiple garments and accessories (e.g. t-shirt, pants, bag, and shoes) within a single denoising pass, while simultaneously unlocking the capability for feature caching during inference.
>
> ### **Q2: Why does the proposed method outperform ReferenceNet with fewer parameters?**
>
> **A2:** This is an excellent question. Intuition might suggest that more parameters (as in ReferenceNet) imply better performance. However, many in-context learning methods, such as CatVTON [1], IC-LoRA [2], and OmniControl [3], have demonstrated that when reference images are utilized as context input within the same model backbone, the network can capture task patterns much more effectively. For instance, CatVTON requires only a few hours of training and approximately 5% of the trainable parameters to converge on the virtual try-on task. This efficiency stems from two critical theoretical and practical reasons:
>
> **1. Feature Space Consistency:** In ReferenceNet, the independent weights of the reference encoder and the main denoising U-Net lead to a misalignment between the reference and denoising feature spaces during attention operations, resulting in performance degradation. In contrast, processing both the reference and denoising inputs within a unified network inherently eliminates this domain gap. This alignment ensures that the model can focus purely on effective feature interaction rather than bridging disparate feature distributions.
>
> **2. Training Stability:** ReferenceNet doubles the parameter count, which increases optimization difficulty. Models with larger parameters are often more sensitive to hyperparameter tuning and are more prone to overfitting. This issue is further exacerbated by public virtual try-on datasets, which typically suffer from limited scale and homogenous data patterns, making the regularization effect of our unified model design even more critical.
>
> We hope these explanations clarify that our method is a distinct architectural advancement.
>
> ### **References:**
>
> [1] Chong, Z., Dong, X., Li, H., Zhang, S., Zhang, W., Zhang, X., Zhao, H., & Liang, X. (2024). CatVTON: Concatenation Is All You Need for Virtual Try-On with Diffusion Models (No. arXiv:2407.15886). arXiv. http://arxiv.org/abs/2407.15886
>
> [2] Huang, L., Wang, W., Wu, Z.-F., Shi, Y., Dou, H., Liang, C., Feng, Y., Liu, Y., & Zhou, J. (2024). In-Context LoRA for Diffusion Transformers (No. arXiv:2410.23775). arXiv. http://arxiv.org/abs/2410.23775
>
> [3] Tan, Z., Liu, S., Yang, X., Xue, Q., & Wang, X. (2024). OminiControl: Minimal and Universal Control for Diffusion Transformer (No. arXiv:2411.15098). arXiv. https://doi.org/10.48550/arXiv.2411.15098

---

### Meta-Review · Area_Chair_BMbU · 2026-01-06

**Summary:**

This paper proposes a diffusion-based framework, i.e., FastFit, for virtual try-on (VTON) technology. The proposed method focuses on two key challenges: the inability of existing methods to simultaneously compose multiple garments/accessories and the computational redundancy caused by re-computing reference garment features at every denoising step. The reviewers have critical concerns on different aspects of this paper, including but not limited to:
1. The technical novelty and contribution are limited. The proposed method to cache reference image information is already widely used in previous methods. Multi-garment try-on has already been explored in prior works such as MMTryon and AnyFit. The main idea seems like a trick rather than a novel algorithm.
2. The proposed method outperforms ReferenceNet with fewer parameters, but the reason is not well explained.
3. Additional evaluation metrics, such as DISTS, would be better for the experimental analysis but are missing in the current form.
4. Visual results are expected but missing corresponding to the ablation studies.
5. Some recent and efficient baselines with strong detail preservation are missing from the comparison, such as ITA-MDT and IMAGDressing-v1.
6. The paper’s clarity requires substantial improvement, as several explanations and figures are confusing or incomplete. The presentation needs a thorough revision.

Some issues are partially addressed. For example, the visual results are provided in the supplementary materials. The major concerns on the technical novelty and the missing comparison with more recent baselines still exist.

**Reviewer Concerns:**

The 3rd, 4th, 5th concerns have been partially addressed, while the 1st, 2nd, and 6th concerns are still outstanding.

**Reviewer Scores:**

None.

---

### Decision · Program_Chairs · 2026-01-26

Reject